# Sequential Ultrasound Assessment of Peri-Articular Soft Tissue in Adhesive Capsulitis of the Shoulder: Correlations with Clinical Impairments—Sequential Ultrasound in Adhesive Capsulitis

**DOI:** 10.3390/diagnostics12092231

**Published:** 2022-09-15

**Authors:** Byung Chan Lee, Seung Mi Yeo, Jong Geol Do, Ji Hye Hwang

**Affiliations:** 1Department of Rehabilitation Medicine, Chungnam National University Hospital, Daejeon 35018, Korea; 2Department of Physical and Rehabilitation Medicine, Pusan National University Yangsan Hospital, Yangsan-si 50612, Korea; 3Department of Physical and Rehabilitation Medicine, Samsung Medical Center, Sungkyunkwan University School of Medicine, Seoul 06351, Korea

**Keywords:** adhesive capsulitis, axillary recess, clinical impairment, ultrasound

## Abstract

Recently, ultrasound measurements of the shoulder such as thickening of the rotator interval (RI) and the axillary recess (AR) are suggested as specific indicators of adhesive capsulitis. Herein, we evaluated the sequential changes in ultrasound parameters and clinical impairments and the correlation between the two in the case of adhesive capsulitis through a prospective observational study of 56 patients with adhesive capsulitis. Clinical assessments and ultrasound parameters, including the thicknesses of the RI and AR, were surveyed at baseline and after 1, 3, and 6 months. In 56 patients with adhesive capsulitis, the thickness of the AR significantly decreased at each follow-up evaluation, but the thickness of the RI showed a significant decrease only between the baseline and 1-month evaluation. In repeated analyses of correlation, the thickness of the AR was strongly correlated with all clinical impairments except the pain at rest and range of internal rotation in the affected shoulder. The thickness of AR was correlated with clinical impairments in patients with adhesive capsulitis during the 6 months follow up and could be useful as a surrogate marker in patients with adhesive capsulitis.

## 1. Introduction

Adhesive capsulitis is a condition that involves pain and gradual loss of range of motion (ROM) in the glenohumeral joint. Fibrosis of the joint capsule and several periarticular structures comprise the pathophysiological features of developing adhesive capsulitis. Among the periarticular structures, the rotator interval (RI), coracohumeral ligament (CHL), and axillary recess (AR) are considered as important anatomical structures [1]. The RI is a triangular-shaped gap between the supraspinatus and subscapularis tendons, and consists of the pulley ligaments (CHL and superior glenohumeral ligament), and intra-articular portion of the biceps tendon [2]. The CHL is a broad ligament of the shoulder that originates from the scapular coracoid process and attaches to the great tuberosity and lesser tuberosity of humerus [1]. Axillary recess is an inferior glenohumeral joint capsule and is located between the anterior and posterior bands of the inferior glenohumeral ligaments [3]. These periarticular structures have a role in stabilizing the shoulder joint; however, they are also associated with other pathologies such as shoulder stiffness or instability.

Recently, several findings specific for adhesive capsulitis have been suggested based on the anatomical structural abnormalities identified through magnetic resonance imaging and ultrasound [1,4,5,6,7,8]. Among them, ultrasound takes the spotlight, owing to its cost-effectiveness, and non-invasive nature. Thickening of the RI, CHL, and AR are specific and reliable findings of adhesive capsulitis [4,8,9,10,11]. Additionally, effusion of the long head of the biceps tendon (LHBT) sheath and hypervascularity in the RI have also been suggested as specific findings of adhesive capsulitis [10,12,13]. Several reports exist on the useful ultrasound parameters that can be used at the time of diagnosis; however, there is no study on the changes in ultrasound parameters in adhesive capsulitis.

Adhesive capsulitis is a self-limiting disease in which recovery occurs after 1–3 years. Treatment options comprise conservative management including physical therapy and intraarticular steroid injection. In refractory cases where conservative treatment has been applied, surgical treatment can be the treatment option [1,14]. However, 20–50% of patients with adhesive capsulitis can develop long-lasting symptoms [7,15]. There are few discussions about why these residual symptoms remain. Therefore, by observing sequential changes in periarticular structures in adhesive capsulitis, the structures that cause these residual symptoms can be localized. However, to the best of our knowledge, no previous studies have reported serial sequential changes of these ultrasound parameters throughout the course of the disease.

The first objective of this study was to evaluate the serial changes of the ultrasound parameters in adhesive capsulitis, and the second objective was to evaluate the associations between the ultrasound parameters and the clinical impairments in patients with unilateral adhesive capsulitis. We hypothesized that the ultrasound parameters change throughout the course of the disease, and that there could be a significant relationship between serial changes of ultrasound features and changes of clinical impairment.

## 2. Materials and Methods

### 2.1. Study Design and Participants

This was a single-arm, prospective, observational study of patients with adhesive capsulitis recruited at a musculoskeletal clinic in a tertiary hospital. The inclusion criteria were as follows: (1) diagnosed with unilateral adhesive capsulitis; (2) age 19 years or older, and; (3) two or more limited ranges of motion of at least 30° in the shoulder joint compared with the contralateral side. The exclusion criteria were as follows: (1) bilateral adhesive capsulitis; (2) secondary adhesive capsulitis; (3) other mimicking disorders such as glenohumeral arthritis, bursitis, rotator cuff disease, and calcific tendinitis of shoulder. After the initial screening, plain X-ray and ultrasound examination were performed to exclude other previously mentioned mimicking disorders. We followed the STROBE guideline; the checklist is attached as Appendix A. Written informed consent was obtained from all patients, and the study protocol was approved by the local ethics committee of Samsung Medical Center (registry number: 2021-12-126).

### 2.2. Baseline Demographics and Clinical Assessments

The medical records of the recruited patients were surveyed for demographic data (age and sex), weight, height, calculated body mass index, and medical history. The clinical outcomes and ultrasound evaluation were assessed at the baseline and at the 1-, 3-, and 6-month follow-ups (Figure 1). The degree of pain was evaluated through the numeric rating scale (NRS) [12]. The passive range of motion (PROM) of the bilateral shoulder was measured in the supine position. Shoulder forward-flexion, abduction, and external/internal rotation at the 90-degree abduction were measured with an electronic goniometer [13]. Shoulder pain and disability experienced in the week prior were assessed using the Shoulder Pain and Disability Index (SPADI) questionnaire [16]. All of the above clinical parameters were surveyed by blinded physical or occupational therapists.

### 2.3. Clinical Management

All patients received hospital-based physiotherapy once or twice a week until the 3-month follow-up and were instructed to exercise at home at least once per day. All exercises consisted of a warm-up, scapular stabilization, shoulder stretching, and cool down exercises. All participants were encouraged and monitored to perform at least 30 min of stepwise educated exercise per day according to the stage of the adhesive capsulitis, including strengthening exercise, from the frozen stage at an outpatient clinic. Participants received an ultrasound-guided intra-articular steroid injection (20 mg of triamcinolone mixed with 5.5 mL of normal saline) using the posterior approach at the initial visit if they had any of the following symptoms: (1) a high intensity of pain (>7 in the NRS scale); (2) consistent night or resting pain; (3) high level of reported disability reported by patients subjectively.

### 2.4. Ultrasound Parameters

Ultrasound examinations were performed on both shoulders with an HM70A (Samsung Medicine, Seoul, Korea) or Volusion E6 (GE Healthcare, Chicago, IL, USA) system, both equipped with a 5–13 MHz linear transducer. The ultrasound measurements of each patient were performed with the same machine throughout the follow-up period. The thickness and vascularity of the RI, thickness of the AR, and effusion of the LHBT were measured in both shoulders. For both the RI and the AR, the ratio of thickness of the affected and unaffected shoulders was calculated. Ultrasonography was performed by three experienced musculoskeletal physiatrists (J.G.D., S.M.Y. and B.C.L.).

### 2.5. Effusion of the LHBT

Each patient was asked to sit and place his or her hand palm up on the knee. The transducer was placed in the axial plane on the body over the anterior shoulder (Figure 2A). The bicipital groove was identified by its characteristic bony contour consisting of greater and lesser tuberosities. The LHBT was located between the greater and lesser tuberosities, within the bicipital groove. Effusion surrounding the LHBT was considered abnormal in a short-axis scan (Figure 2B).

### 2.6. RI Thickness and Hypervascularity

The participants were asked to sit with their shoulder extended, elbow flexed, and forearm supinated. The practitioners placed the transducer on the supraspinatus tendon in a long axis view and then rotated it by 90° to obtain a short axis view (Figure 2C). The transducers were distally moved to visualize the LHBT situated between the supraspinatus and subscapularis tendons [17]. The thin, hyperechoic, soft tissues were identified as superficially placed to the LHBT (Figure 2D). A B-mode ultrasound and power Doppler were performed at the above-described view. The thickness of the soft tissue in the RI was measured as the shortest distance between the LHBT and peribursal fat. Hypervascularity was measured as either absent or present by a high signal within the soft tissues of RI in the power Doppler measurement.

### 2.7. AR Thickness

The participants were asked to lie supine with the shoulder abducted, elbow flexed, and forearm neutral. The ultrasound probe was placed on the mid-axillary line longitudinally along the shaft of the humerus (Figure 2E). A caliper on the ultrasound machine was used to measure the AR thickness as the distance from the cortical line of the humerus to the outer margin of the glenohumeral joint capsule at the humeral neck in a still image, which indicated the cortical line of the humerus (Figure 2F) [10].

### 2.8. Statistical Analysis

Descriptive statistics were used to characterize the demographic and clinical variables. Continuous variables are presented as means and standard deviations. The Mann-Whitney test or independent T-test was used to compare the thicknesses of RI and AR and Fisher’s exact test was used to compare the frequencies of effusion of the LHBT and increased vascularity of the RI between the affected and unaffected shoulders. The paired *t*-test or the Wilcoxon signed rank test was employed to evaluate the sequential changes of the ultrasound parameters and clinical impairment scores. Missing data from patients with a waived consent were replaced using a mean imputation only in evaluating the sequential changes of the ultrasound parameters and clinical impairments. The Bonferroni correction was used to adjust the *p*-values in evaluating sequential changes of clinical variables and ultrasound parameters.

A repeated measures correlation analysis was used to evaluate the correlations between ultrasound parameters and clinical impairments [18]. The correlation effect size was classified according to the correlation coefficient with a value of 0.10–0.30 classified as a weak association, 0.30–0.50 as a moderate association, and 0.50 or larger as a strong association. Data were analyzed using R version 4.1.1. All statistical tests were two-sided at a 5% significance level.

## 3. Results

### 3.1. Patient Characteristics

A total of 56 participants (49 women, 7 men) was enrolled. During follow-up, 13 patients were excluded due to a withdrawal of consent (Figure 1). The mean age of the 56 patients was 53.1 ± 7.7 years. Four patients (7.1%) had diabetes, 32 (57.1%) were affected with dominant extremity, and 41 (73.2%) received a corticosteroid injection. Mean NRS scores of the initial pain intensity at rest and during activity were 1.9 ± 2.1 and 5.8 ± 2.0, respectively. The mean total, disability and pain sections percentage scores of SPADI were 43.0 ± 17.3%, 46.7 ± 17.4% and 40.7 ± 20.3%, respectively. The initial PROM of the affected shoulder was 124.5 ± 18.1° in forward flexion, 87.2 ± 16.7° in abduction, 34.0 ± 16.4° in internal rotation, and 33.0 ± 14.1° in external rotation (Table 1).

### 3.2. Initial Values and Sequential Changes of Ultrasound Parameters

The thicknesses of the RI (2.0 ± 0.6 mm vs. 1.7 ± 0.5 mm, *p* = 0.026) and AR (4.6 ± 1.6 mm vs. 2.5 ± 0.7 mm, *p <* 0.001) were significantly greater in the affected shoulder than those in the unaffected shoulder. Effusion of the LHBT sheath was significantly more frequent in the affected shoulder (35.7% vs. 5.4%, *p <* 0.001). However, the increased vascularity of RI were not significantly different among the two groups (1.8% vs. 0%, *p* = 1.000) (Table 2).

The thickness of AR changed significantly (baseline: 4.6 ± 1.6 mm; 1-month: 4.0 ± 1.3 mm; 3-month: 3.4 ± 1.1 mm; 6-month: 3.0 ± 0.9 mm) (Figure 3 and Figure 4), while the ratio of AR thickness also changed significantly (baseline: 191.5 ± 63.2%; 1-month: 169.6 ± 50.5%; 3-month: 152.0 ± 38.9%; 6-month: 133.7 ± 34.2%). The thickness of the RI also changed; however, significant changes were observed only between the baseline and the 1-month follow-up evaluation (baseline: 2.0 ± 0.6 mm; 1-month: 1.8 ± 0.5 mm; 3-month: 1.8 ± 0.5 mm; 6-month: 1.7 ± 0.4 mm). The affected/unaffected ratio of RI thickness had no statistically significant changes (baseline: 115.4 ± 28.0%; 1-month: 102.1 ± 20.0%; 3-month: 104.6 ± 17.6%; 6-month: 98.8 ± 14.6%) (Figure 4).

### 3.3. Sequential Changes in Clinical Impairments

Significant improvements were observed in all the measured clinical impairments in participants. Pain intensity at rest and during activity improved between the initial visit and at the final follow-up. Clinical impairment, measured by SPADI, also improved in the total, pain, and disability sections. Limited ROM in every direction of movement was also improved between the initial visit and final follow-up (Table 3).

### 3.4. Correlations between UItrasound Parameters and Clinical Variables

A repeated measures correlation analysis revealed that the AR thickness and its affected/unaffected ratio were significantly and strongly correlated with pain, SPADI score, and PROM except for pain in the resting state and the internal rotation of the affected shoulder. The thickness of the RI and its ratio showed a weak or moderate correlation with clinical impairment and did not have a strong correlation with any of the clinical measurements (Table 4).

## 4. Discussion

This study evaluated the serial ultrasound findings and their association with the corresponding clinical impairments in 56 patients with adhesive capsulitis. In the sequential ultrasonography measurements, the thickness of the AR significantly decreased at each follow-up, but the thickness of RI showed a significant decrease only between the baseline and 1-month evaluation. Clinical impairments had a significant association with the measured ultrasound parameters. The thickness of the AR was strongly correlated with the clinical impairments except for the NRS score for the pain at rest and range of internal rotation in the affected shoulder. The thickness of the RI showed weak to moderate correlations with the clinical impairments.

Inflammation and sequential fibrotic constriction of the periarticular and joint structures are potential causes of disability in adhesive capsulitis. Several reports demonstrated abnormalities in these structures in adhesive capsulitis, including thickening of the CHL, AR and RI, enhancement in the AR and obliteration of the fat triangle below the coracoid process in MRI analyses [7,19,20]. Ultrasound measurements also demonstrated a thickening of the RI, CHL, and AR in patients with adhesive capsulitis [4,8,9,10,11]. Effusion of the LHBT sheath and hypervascularity in the RI have also been suggested as specific findings related to adhesive capsulitis [9,21,22]. Recently, a reduced sliding of the infraspinatus tendon, called bounce sign, was also suggested as a potential diagnostic tool for adhesive capsulitis (sensitivity 41.5%, specificity 100%) [8]. The mean value of the thickness of the AR measured at baseline differed significantly between the affected (4.6 ± 1.7 mm) and unaffected shoulders (2.5 ± 0.7 mm) in this study. At the baseline evaluation, the mean thickness of the RI (2.0 ± 0.5 mm) in the affected shoulders was significantly greater than that in the unaffected shoulders (1.7 ± 0.5 mm). Effusion of the LHBT on the side with adhesive capsulitis was more frequently observed than that in the contralateral limb. Those findings were consistent with the previous studies. Effusion of the LHBT is related to intra-articular pathologies of shoulder because the tendon sheath is derived from the extension of the glenohumeral joint capsule. Therefore, an increased effusion of the glenohumeral joint exacerbates the effusion of the LHBT [23]. However, based on the results of this study, an increased vascularity in the RI was not a reliable sign of adhesive capsulitis.

Several previous reports have suggested a correlation between the anatomical imaging parameters and clinical variables [4,24,25]. Park et al. [25] reported that the joint capsular edema at the AR was significantly correlated with the ROM on external rotation, and the joint capsule thickness decreased depending on the stage of adhesive capsulitis (stage 1: mean 4.7 mm; stage 2: 3.7 mm; stage 3: 3.7 mm). Kanazawa et al. [24] also reported significant negative correlations between the thickness of the joint capsule on the glenoid side and external rotation and placement of the hand behind the back on MRI. Do et al. [4] reported that the thickness of the AR was significantly correlated with the forward elevation (R = −0.28, *p* = 0.028) and internal rotation (R = 0.456, *p* < 0.001) movements. However, the studies above were cross-sectional, and serial changes in these structural parameters were not assessed. Defining sequential changes in these periarticular structures could provide a possible explanation for chronic disability in patients with adhesive capsulitis.

In our study, chronological changes of the AR and RI were observed through serial ultrasound measurements. The thickness of the AR and its affected/unaffected ratio changed significantly throughout the follow-up period. The thickness of the AR showed a strong correlation with all clinical impairments except the internal rotation of the affected shoulder and NRS score of pain at rest. Moreover, 6 months after the baseline evaluation, the thickness of the AR in the affected shoulder had not decreased to that of the unaffected shoulder, with a mean affected/unaffected ratio of 133.7 ± 39.2%. These results suggest that the reversal of fibrotic changes in joint capsules takes more than 6 months. The thickness of the RI changed significantly from baseline to the 1-month follow-up evaluation and did not significantly change in the subsequent follow-up evaluations and showed a significant but not strong correlation with the clinical impairments. These results imply that the thickness of the RI could be the result of inflammation rather than the fibrosis of periarticular structures.

The natural history of adhesive capsulitis is controversial. Vastamake et al. [26] have reported that 94% of the 84 patients with adhesive capsulitis returned to their normal function and ROM during a follow-up of 2–24 years. However, Hand et al. [27] have shown that 41% of 223 patients had residual complaints during a follow-up of 2–20 years. Additionally, Kim et al. [7] reported that 20.1% of 234 patients had residual shoulder pain and 39.7% of participants had persistent motion deficits after 2 years of follow-up. These discrepancies between studies could be the result of uncertainty of the pathophysiology and the natural course of adhesive capsulitis.

The pathophysiology of adhesive capsulitis has not been clearly elucidated; however, inflammatory contracture of the shoulder joint could be considered its primary histopathology. These pathologic findings have been confirmed by the presence of increased fibroblasts mixed with type I and III collagen in the joint capsule and synovium, suggesting adhesive capsulitis as a fibrotic disorder [28,29]. Several recent studies have also demonstrated several elevated inflammatory cytokines in the capsular and bursal tissues of patients with adhesive capsulitis, including interleukin-1a, interleukin-1b, tumor necrosis factor-a, cyclooxygenase-1, and cyclooxygenase-2 [30]. Furthermore, an imbalance between metalloproteinase and tissue inhibitors of metalloproteinase has been demonstrated in previous reports [31]. Therefore, inflammatory processes and fibrotic changes occur concurrently in adhesive capsulitis of the shoulder. These fibrotic changes in the capsular or pericapsular structure could be a possible explanation for the limitation of glenohumeral movement. Therefore, based on the results of our study, fibrotic changes that occur in the AR are typically reversed, and this reversal process requires more than 6 months. In patients with residual deficits, evaluating the inferior glenohumeral ligament and joint capsules in the AR may provide a possible explanation of the residual deficit and can suggest potential treatment options for physicians to treat adhesive capsulitis.

This study has several limitations. First, ultrasound is an experience-dependent method, and measurement values differ by operator and ultrasound instrument. However, all three measurers were experienced in musculoskeletal and ultrasound examination for more than 5 years in the same facility. Additionally, in this study, to attempt to eliminate the observed discrepancy, regular discussions of the protocol of ultrasound measurement and regular meetings before and between ultrasound evaluations were conducted. Second, external and internal rotation ROM in glenohumeral joint were measured with patients at 90-degree abduction in supine position. The authors decided to apply those measurement methods to control the scapular motions in participants. However, external rotation at the side and hand-behind-back methods could be most popular methods to evaluate the limited glenohumeral joint motions in clinics. Therefore, those measurement differences could cause potential bias. Third, participants in this study were not entirely representative of the general population of patients with adhesive capsulitis and the natural course of adhesive capsulitis because of the imbalance between the characteristics of male and female patients, as well as the use of intra-articular steroid injection. A long-term, large, and general group observational analysis of patients with adhesive capsulitis is needed to validate the results in this study.

## 5. Conclusions

Sequential ultrasound measurements in patients with adhesive capsulitis showed that the thickness of the AR and RI changed with time, and that clinical parameters of pain, SPADI score, and PROM in the affected shoulder were recovered in the follow-up period and had a significant association with the ultrasound measurements, especially the thickness of AR. To the best of our knowledge, this is the first prospective observational study on the sequential changes in anatomical structures measured by ultrasound. Further large-scale, longer duration observational studies are needed.

## Figures and Tables

**Figure 1 diagnostics-12-02231-f001:**
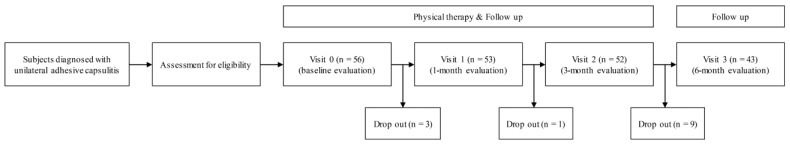
Flow chart of the single-arm, prospective study.

**Figure 2 diagnostics-12-02231-f002:**
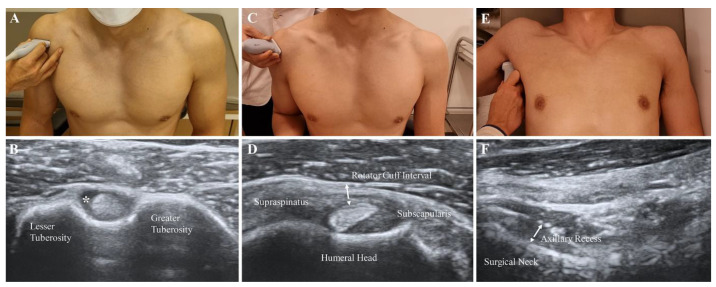
Measurement of features with B-mode ultrasound scan. (**A**,**B**) Effusion of the long head of the biceps tendon sheath (asterisk). (**C**,**D**) Oblique axial view of the soft tissue in the rotator cuff interval (dotted arrow). (**E**,**F**) Thickness of the axillary recess at the humeral surgical neck (dotted arrow).

**Figure 3 diagnostics-12-02231-f003:**
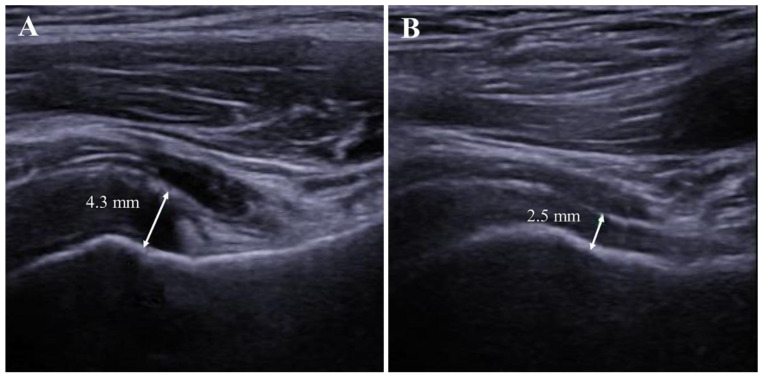
Follow-up evaluation of axillary recess in the affected shoulder. (**A**). Thickened axillary recess measured at 4.3 mm at the initial (baseline) visit. (**B**). Follow-up evaluation of the same patient showed a decreased thickness of the axillary recess after 6 months.

**Figure 4 diagnostics-12-02231-f004:**
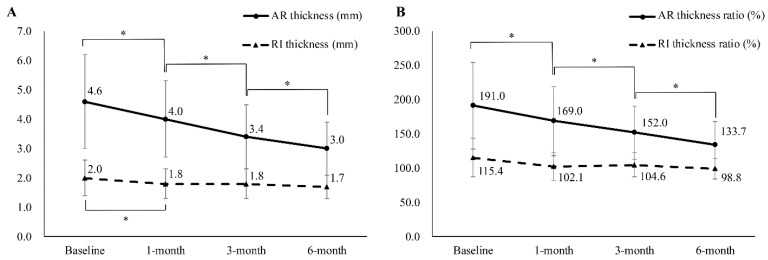
(**A**) Sequential changes in the ultrasound parameters of patients with adhesive capsulitis. (**B**) Sequential changes of ratio of ultrasound parameters in the same population. The ratio between the thicknesses of the RI and the AR of the affected and unaffected shoulders was calculated by dividing the value of the affected shoulder by that of the unaffected shoulder and was converted into a percentage score. AR, axillary recess; RI, rotator interval. * Significantly different from previous evaluation.

**Table 1 diagnostics-12-02231-t001:** Epidemiologic and clinical characteristics of the patients at baseline evaluation.

	Total (*n*= 56)
Age, year	53.1 ± 7.7
Female, n (%)	49 (87.5%)
DM, n (%)	4 (7.1%)
Dominant side affected, n (%)	32 (57.1%)
Corticosteroid injection	41 (73.2%)
BMI (kg/m^2^)	17.2 ± 3.3
NRS (pain at resting)	1.9 ± 2.1
NRS (pain at activity)	5.8 ± 2.0
SPADI, pain (%)	40.7 ± 20.3
SPADI, disability (%)	46.7 ± 17.4
SPADI, total (%)	43.0 ± 17.3
FE, deg	124.5 ± 18.1
Abduction, deg	87.2 ± 16.7
IR, deg	34.0 ± 16.4
ER, deg	33.0 ± 14.1

Values are expressed as the mean ± SD or n (%). BMI: body mass index; DM: diabetes mellitus; ER: external rotation; FE: forward elevation; IR: internal rotation; NRS: numeric rating scale; ROM: range of motion; SPADI: shoulder pain and disability index.

**Table 2 diagnostics-12-02231-t002:** Ultrasound parameters between affected and unaffected shoulders at baseline evaluation.

*n* = 56	Affected	Unaffected	*p*-Value
AR thickness, mm	4.6 ± 1.6	2.5 ± 0.7	<0.001 *
RI thickness, mm	2.0 ± 0.6	1.7 ± 0.5	0.026 *
LHBT effusion	20 (35.7%)	3 (5.4%)	<0.001 *
Hypervascularity of RI	1 (1.8%)	0 (0%)	1.000

Values are expressed as the mean ± SD or n (%). * *p* < 0.05 between affected and unaffected shoulders. AR: axillary recess; LHBT: long head of the biceps tendon; RI: rotator interval.

**Table 3 diagnostics-12-02231-t003:** Sequential changes in the clinical outcomes of patients with adhesive capsulitis.

	Initial	1-Month	3-Month	6-Month
NRS, pain (resting)	1.9 ± 2.1	0.3 ± 0.7 *	0.2 ± 0.5	0.1 ± 0.4
NRS, pain (activity)	5.8 ± 2.0	1.9 ± 1.1 *	0.8 ± 1.1 ^†^	0.6 ± 0.9
SPADI, pain (%)	46.7 ± 17.4	17.3 ± 10.7 *	10.1 ± 8.1 ^†^	7.2 ± 7.0
SPADI, disability (%)	40.7 ± 20.3	13.3 ± 10.3 *	5.4 ± 5.4 ^†^	3.2 ± 5.6 ^‡^
SPADI, total (%)	43.0 ± 17.3	14.8 ± 9.1 *	7.3 ± 5.6 ^†^	4.8 ± 5.9 ^‡^
FE, deg	124.5 ± 18.1	152.7 ± 13.6 *	168.1 ± 10.6 ^†^	172.1 ± 11.6 ^‡^
Abduction, deg	87.2 ± 16.7	119.9 ± 17.8 *	151.2 ± 24.3 ^†^	162.6 ± 22.3 ^‡^
IR, deg	34.0 ± 16.4	51.8 ± 14.1 *	61.2 ± 11.9 ^†^	62.0 ± 11.2
ER, deg	33.0 ± 14.1	57.3 ± 15.3 *	72.1 ± 15.7 ^†^	76.0 ± 15.2

Values are expressed as mean ± SD. ER: external rotation; FE: forward elevation; IR: internal rotation; NRS: numeric rating scale; ROM: range of motion; SPADI: shoulder pain and disability index. * Significantly different from baseline, ^†^ Significantly different from 1-month measurement, ^‡^ Significantly different from 3-month measurement.

**Table 4 diagnostics-12-02231-t004:** Repeated measures correlation between ultrasound parameters and clinical variables.

	ART	ART Ratio	RIT	RIT Ratio
NRS, pain (resting)	R: 0.363	R: 0.376	R: 0.241	R: 0.406
	*p* < 0.001	*p* < 0.001	*p* = 0.003	*p* < 0.001
NRS, pain (activity)	R: 0.557	R: 0.539	R: 0.359	R: 0.391
	*p* < 0.001	*p* < 0.001	*p* < 0.001	*p* < 0.001
SPADI pain (%)	R: 0.580	R: 0.578	R: 0.351	R: 0.450
	*p* < 0.001	*p* < 0.001	*p* < 0.001	*p* < 0.001
SPADI, disability (%)	R: 0.540	R: 0.577	R: 0.330	R: 0.467
	*p* < 0.001	*p* < 0.001	*p* < 0.001	*p* < 0.001
SPADI total (%)	R: 0.569	R: 0.591	R: 0.346	R: 0.471
	*p* < 0.001	*p* < 0.001	*p* < 0.001	*p* < 0.001
FE	R: −0.584	R: −0.594	R: −0.333	R: −0.389
	*p* < 0.001	*p* < 0.001	*p* < 0.001	*p* < 0.001
AB	R: −0.566	R: −0.554	R: −0.318	R: −0.330
	*p* < 0.001	*p* < 0.001	*p* < 0.001	*p* < 0.001
IR	R: −0.436	R: −0.434	R: −0.228	R: −0.278
	*p* < 0.001	*p* < 0.001	*p* = 0.005	*p* < 0.001
ER	R: −0.571	R: −0.538	R: −0.340	R: −0.339
	*p* < 0.001	*p* < 0.001	*p* < 0.001	*p* < 0.001

AB: abduction; ART: axillary recess thickness; ER: external rotation; FE: forward elevation; IR: internal rotation; NRS: numeric rating scale; RIT: rotator interval thickness; SPADI: shoulder pain and disability index.

## Data Availability

The data presented in this study are available on request from the corresponding author.

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
