# Peer review of "Sequential Ultrasound Assessment of Peri-Articular Soft Tissue in Adhesive Capsulitis of the Shoulder: Correlations with Clinical Impairments—Sequential Ultrasound in Adhesive Capsulitis"

_diagnostics, 2022, doi:10.3390/diagnostics12092231_

Round 1

Reviewer 1 Report

The study is well written and results are reported clearly. However, a sample size calculation is missing. The STROBE guidelines for reporting observational studies should be implemented and the checklist included. We suggest revision by a native English for grammar and typos. 

In addition, we suggest to discuss the recent evidence published on Rheumatology and Therapy reporting novel ultrasound parameters for adhesive capsulitis diagnosis Stella, S.M., Gualtierotti, R., Ciampi, B. et al. Ultrasound Features of Adhesive Capsulitis. Rheumatol Ther 9, 481–495 (2022). https://doi.org/10.1007/s40744-021-00413-w. In particular, an explanation should be added regarding the fact that the rotator interval is composed of two ligaments (the coracohumeral ligament and the superior glenohumeral ligament).

Author Response

Response to Reviewer 1 Comments

We appreciate the reviewer’s positive recommendation of our work and the valuable comments for further improving the paper. Our item-by-item response is as follows, and the corresponding corrected text in the manuscript is highlighted. Also, please see the attachment files. 

Point 1: The study is well written, and results are reported clearly. However, a sample size calculation is missing.

Response 1: Thanks for the comment. However, since the design of our study is a single-arm, prospective, observational case series of patients with adhesive capsulitis, authors think that calculating the sample size in this study is not appropriate.

Point 2: The STROBE guidelines for reporting observational studies should be implemented and the checklist included.

Response 2: Thanks for the comments. We decided to implement the STROBE guidelines for our study and have attached the relevant details as supplemental material. We also added comment in the material and methods section as follows:

P2 line 92-93: We followed the STROBE guideline, and the checklist is attached as Table S1.

Point 3: We suggest revision by a native English for grammar and typos.

Response 3: Thanks for the comments. We received proofreading of our manuscript by a native English Speaker and have attached the certificate for proof reading.

Point 4: In addition, we suggest to discuss the recent evidence published on Rheumatology and Therapy reporting novel ultrasound parameters for adhesive capsulitis diagnosis Stella, S.M., Gualtierotti, R., Ciampi, B. et al. Ultrasound Features of Adhesive Capsulitis. Rheumatol Ther 9, 481–495 (2022). https://doi.org/10.1007/s40744-021-00413-w. In particular, an explanation should be added regarding the fact that the rotator interval is composed of two ligaments (the coracohumeral ligament and the superior glenohumeral ligament).

Response 4: Thanks for the comments. We have thoroughly read your recommended reference, and have added the following text in the introduction and material and methods sections as follows and quote the recommended articles in reference 9:

P2 line 44-56; Fibrosis of joint capsule and several periarticular structures comprise the pathophysiology of developing adhesive capsulitis. Among the periarticular structures, the rotator interval (RI), coracohumeral ligament (CHL), and axillary recess (AR) are considered the important anatomical structures [1]. The RI is a triangular shaped gap between supraspinatus and subscapularis tendons. The RI consists of the pulley ligaments (CHL and superior glenohumeral ligament), and intra-articular portion of the biceps tendon [2]. The CHL is a broad ligament of the shoulder that originates from the scapular coracoid process and attaches to the great tuberosity and lesser tuberosity of the humerus [1]. Axillary recess is an inferior glenohumeral joint capsule and located between the anterior and posterior bands of the inferior glenohumeral ligaments [3]. These periarticular structures have a role in stabilizing the shoulder joint; however, they are also associated with other pathologies such as shoulder stiffness or instability.

P8 line 265-267; Recently, reduced sliding of the infraspinatus tendon, called bounce sign, was also suggested as a potential diagnostic tool of adhesive capsulitis (sensitivity 41.5%, specificity 100%) [9].

Reviewer 2 Report

Many thanks to the authors for having presented a so interesting external validation study about “Sequential Ultrasound Assessment of Peri-articular Soft Tissue in Adhesive Capsulitis of the Shoulder: Correlations with Clinical Impairments”. The language is so good that the manuscript does not need to be corrected by a person of English mother tongue.

Abstract

The abstract is well structured and it contains the main results of the study, however I would personally add the possible role of RI as a parameter in the study of Adhesive Capsulitis of the Shoulder as done for RA in Lines 27-29, plus in this paragraph there is no mention of Effusion of the LHBT, I would personally add that.

Please, integrate possible treatment methods for capsulitis, including surgical ones, and quote:

·         Mid-Term Outcomes after Arthroscopic "Tear Completion Repair" of Partial Thickness Rotator Cuff Tears. Medicina (Kaunas) 2021 Jan 17;57(1):74. doi: 10.3390/medicina57010074.

Moreover, keywords should be sorted in alphabetic order.

Background

The introduction is well structured, the rationale behind the study is written in a clear and understandable way, moreover it includes the main aims of the study.

I would personally add a brief definition of the rotator interval (RI) and the axillary recess (AR) and the role of their thickening in the development/aggravation of Adhesive Capsulitis in the Shoulder, plus a brief definition of the Effusion of the LHBT and its role in the development/aggravation of Adhesive Capsulitis in the Shoulder.

Methods

This section contains enough information to understand and possibly repeat the study, every aspect of the study has been considered and explained in detail. I would personally add some information regarding the correlation, if that is the case, between hand dominance and shoulder affection in the patients studied, moreover I would add some information regarding the degree of physical activity implemented by the patients of the study. 

 Results

The results presented are quite complete, reflecting the M&M section. The tables are comprehensive and when integrated into the text they provide a complete and understandable picture of the results.

Regarding the tables, the abbreviations’ extensions should be sorted in alphabetical order.

 Discussion

The length and content of the discussion communicate the main information of the paper. Also, the authors provided several limitations of this study, furthermore adding future potential studies and implementations necessary to further clarify the matter.   

Conclusion
The conclusions reflect and refer to the results of the study. It is written in a schematic way and it focuses the matters of the study, plus briefly providing limitations of the study.

References

The references are up to date. Hence, delate those before 2010 if not essential (nr. 3, 7, 10, 13, 14, 16, 19, 25, 26, 27) replacing them with newer ones and integrate them as suggested previously.

Tables and Figures

The number and quality of tables and figures are appropriate to transmit the main information of the paper.

Author Response

Response to Reviewer 2 Comments 

We appreciate the reviewer’s positive recommendation of our work and the valuable comments for further improving the paper. Our item-by-item response is as follows, and the corresponding corrected text in the manuscript is highlighted. Please see the attachment.

Point 1: The abstract is well structured and it contains the main results of the study, however I would personally add the possible role of RI as a parameter in the study of Adhesive Capsulitis of the Shoulder as done for AR in Lines 27-29, plus in this paragraph there is no mention of Effusion of the LHBT, I would personally add that.

Response 1: Thanks for the comment. We have considered the use of RI as a parameter in the study of adhesive capsulitis and added the description in the Abstract. However, Authors guidelines in ‘Diagnostics’ journal stated specifically that “The abstract should be a total of about 200 words maximum.

Therefore, due to the limitation of the length of abstract (currently, 190 words), we decided not to include the potential role of the RI and the effusion of the LHBT in the abstract.

 We also consider that the key point of our study is the chronological change in the ultrasound parameters and its potential use as a surrogate marker in clinical practice. Therefore, we decided to emphasize that the potential role of AR in sequential ultrasound is more important than that of the RI and effusion in the LHBT.

Point 2: Please, integrate possible treatment methods for capsulitis, including surgical ones, and quote:

  • Mid-Term Outcomes after Arthroscopic "Tear Completion Repair" of Partial Thickness Rotator Cuff Tears. Medicina (Kaunas) 2021 Jan 17;57(1):74. doi: 10.3390/medicina57010074.

Response 2: Thanks for the comment. We have briefly described the treatment options of adhesive capsulitis in the manuscript as follows and quote the recommended articles in reference 16:

P2 line 67-69; Treatment options comprise conservative management including physical therapy and intraarticular steroid injection. In refractory cases where conservative treatment has been applied, surgical treatment can be the treatment option [1,16].

Point 3: Moreover, keywords should be sorted in alphabetic order.

 Response 3: Thanks for the comments. Authors sorted the Keywords and list of abbreviations in alphabetical order and highlighted in the manuscript.

Keywords: Adhesive capsulitis; Axillary recess; Clinical impairment; Ultrasound

List of Abbreviation

AR, Axillary Recess; CHL, Coracohumeral Ligament; LHBT, Long Head of the Biceps Tendon; NRS, Numeric Rating Scale; PROM, Passive Range of Motion; SPADI, Shoulder Pain and Disability Index; RI, Rotator Interval; ROM, Range of motion

Point 4: The introduction is well structured, the rationale behind the study is written in a clear and understandable way, moreover it includes the main aims of the study.

I would personally add a brief definition of the rotator interval (RI) and the axillary recess (AR) and the role of their thickening in the development/aggravation of Adhesive Capsulitis in the Shoulder, plus a brief definition of the Effusion of the LHBT and its role in the development/aggravation of Adhesive Capsulitis in the Shoulder.

 Response 4: Thanks for the comment. Authors have added a brief definition of the rotator interval, axillary recess, and coracohumeral ligaments in the Introduction section as follows. Also, we have added the meaning of the effusion of the long head of the biceps tendon in adhesive capsulitis in the shoulder as follows briefly.

P2 line 44-56; Fibrosis of the joint capsule and several periarticular structures comprise the pathophysiology of developing adhesive capsulitis. Among the periarticular structures, the rotator interval (RI), coracohumeral ligament (CHL), and axillary recess (AR) are considered as the important anatomical structures [1]. The RI is a triangular shaped gap be-tween supraspinatus and subscapularis tendons. The RI consists of the pulley ligaments (CHL and superior glenohumeral ligament), and intra-articular portion of the biceps tendon [2]. The CHL is a broad ligament of the shoulder that originates from the scapular coracoid process and attaches to the great tuberosity and lesser tuberosity of the humerus [1]. Axillary recess is an inferior glenohumeral joint capsule and is located between the anterior and posterior bands of the inferior glenohumeral ligaments [3]. These periarticular structures have a role in stabilizing the shoulder joint; however, they are also associated with other pathologies such as shoulder stiffness or instability.

P2 line 57 – 59;

Before: Recently, several findings specific for adhesive capsulitis have been suggested based on magnetic resonance imaging and ultrasound

After: Recently, several findings specific for adhesive capsulitis have been suggested based on these anatomical structural abnormalities identified through magnetic resonance imaging and ultrasound [4-9].

P8-9 line 273-276; Effusion of the LHBT is related to intra-articular pathologies of shoulder because the tendon sheath is derived from the extension of the glenohumeral joint capsule. Therefore, an increased effusion of the glenohumeral joint exacerbates the effusion of the LHBT [24].

Point 5: This section contains enough information to understand and possibly repeat the study, every aspect of the study has been considered and explained in detail. I would personally add some information regarding the correlation, if that is the case, between hand dominance and shoulder affection in the patients studied, moreover I would add some information regarding the degree of physical activity implemented by the patients of the study. 

 Response 5:

Thanks for the kind comment. We have analyzed the relevance between hand dominance and shoulder affected side. The result of the analysis is attached as a table as bellow. Any statistical significance was not observed, and authors decided to exclude these results in the manuscript because those results does not match with our focus of the study.

Right Affected

Left affected

Total

P-values

Right dominance

30

22

52 (92.9%)

1.000

Left dominance

2

2

4 (7.1%)

Total

32 (57.1%)

24 (42.9%)

Physical activities of the participants were described in the manuscript as follows.

P3 line 111-114; All patients received hospital-based physiotherapy once or twice a week until the 3-month follow-up and were instructed to exercise at home at least once per day. All exercises consisted of a warm-up, scapular stabilization, shoulder stretching, and cool down exercises.

However, as reviewer pointed, we have updated the manuscript including the details on the degree of exercises. Authors instruct the progressive stepwise exercise according to the stage of adhesive capsulitis in patients including strengthening exercise from the frozen stage.  

P3 line 114-116; All participants were encouraged and monitored to perform at least 30 minutes of stepwise educated exercise per day according to the stage of the adhesive capsulitis, including strengthening exercise, from the frozen stage at an outpatient clinic.

Point 6: The results presented are quite complete, reflecting the M&M section. The tables are comprehensive and when integrated into the text they provide a complete and understandable picture of the results. Regarding the tables, the abbreviations’ extensions should be sorted in alphabetical order.

Response 6: Thanks for the comments. Authors have sorted the abbreviation in alphabetical order and have highlighted them in the manuscript as follows.

Table 1. Epidemiologic and clinical characteristics of the patients at baseline evaluation.

BMI: body mass index; DM: diabetes mellitus; ER: external rotation; FE: forward elevation; IR: internal rotation; NRS: numeric rating scale; ROM: range of motion; SPADI: shoulder pain and disability index.

Table 2. Ultrasound parameters between affected and unaffected shoulders at baseline evaluation

AR: axillary recess; LHBT: long head of the biceps tendon; RI: rotator interval.

Table 3. Sequential changes in the clinical outcomes of patients with adhesive capsulitis.

ER: external rotation; FE: forward elevation; IR: internal rotation; NRS: numeric rating scale; ROM: range of motion; SPADI: shoulder pain and disability index.

Table 4. Repeated measures correlation between ultrasound parameters and clinical variables.

AB: abduction; ART: axillary recess thickness; ER: external rotation; FE: forward eleva-tion; IR: internal rotation; NRS: numeric rating scale; RIT: rotator interval thickness; SPADI: shoulder pain and disability index.

Point 7: The length and content of the discussion communicate the main information of the paper. Also, the authors provided several limitations of this study, furthermore, adding future potential studies and implementations necessary to further clarify the matter.   

Response 7: Thanks for the kind comments.

Point 8: The conclusions reflect and refer to the results of the study. It is written in a schematic way and it focuses the matters of the study, plus briefly providing limitations of the study.

Response 8: Thanks for the kind comments.

Point 9: The references are up to date. Hence, delate those before 2010 if not essential (nr. 3, 7, 10, 13, 14, 16, 19, 25, 26, 27) replacing them with newer ones and integrate them as suggested previously.

Response 9: Thanks for the kind comments.

As you recommend, we replace the reference before 2010 with newer ones as follows

Before:  

  1. Mengiardi, B.; Pfirrmann, C.W.; Gerber, C.; Hodler, J.; Zanetti, M. Frozen shoulder: MR arthrographic findings. Radiology 2004, 233, 486-492.

After:

  1. Choi, Y.H.; Kim, D.H. Correlations between clinical features and MRI findings in early adhesive capsulitis of the shoulder: a retrospective observational study. BMC musculoskeletal disorders 2020, 21, 542.

Before:

  1. Homsi, C.; Bordalo-Rodrigues, M.; da Silva, J.J.; Stump, X.M. Ultrasound in adhesive capsulitis of the shoulder: is assessment of the coracohumeral ligament a valuable diagnostic tool? Skeletal Radiol. 2006, 35, 673-678.

After:

  1. Stella, S.M.; Gualtierotti, R.; Ciampi, B.; Trentanni, C.; Sconfienza, L.M.; Del Chiaro, A.; Pacini, P.; Miccoli, M.; Galletti, S. Ultrasound Features of Adhesive Capsulitis. Rheumatology and Therapy 2022, 9, 481-495.

Authos also decided to delete the following reference because the other included references provided enough

explanation of the content of the manuscript. 

  1. Emig, E.W.; Schweitzer, M.E.; Karasick, D.; Lubowitz, J. Adhesive capsulitis of the shoulder: MR diagnosis. AJR Am. J. Roentgenol. 1995, 164, 1457-1459.

However, the authors decided not to delete or replace the references mentioned below because these refrences are not replaceable. Reference 10 describes the hypervacularity of rotator interval in the ultrasound, reference 13, 25 includes the study of long term sequelae of adhesive capsulitis. Reference 14, 16 states the original description of the numeric rating scale and shoulder pain and disability index. Reference 26 and 27 include the basic science articles to elucidate the pathophysiology of the adhesive capsulitis. Therefore, those articles are mandantory to describe the results of our sutdy.

  1. Lee, J.C.; Sykes, C.; Saifuddin, A.; Connell, D. Adhesive capsulitis: sonographic changes in the rotator cuff interval with arthroscopic correlation. Skeletal Radiol. 2005, 34, 522-527.
  2. Shaffer, B.; Tibone, J.E.; Kerlan, R.K. Frozen shoulder. A long-term follow-up. J. Bone Joint Surg. Am. 1992, 74, 738-746.
  3. Hartrick, C.T.; Kovan, J.P.; Shapiro, S. The numeric rating scale for clinical pain measurement: a ratio measure? Pain Pract. 2003, 3, 310-316.
  4. Roach, K.E.; Budiman-Mak, E.; Songsiridej, N.; Lertratanakul, Y. Development of a shoulder pain and disability index. Arthritis Care Res. 1991, 4, 143-149.
  5. Hand, C.; Clipsham, K.; Rees, J.L.; Carr, A.J. Long-term outcome of frozen shoulder. J. Shoulder Elbow Surg. 2008, 17, 231-236.
  6. Bunker, T.D.; Anthony, P.P. The pathology of frozen shoulder. A Dupuytren-like disease. J. Bone Joint Surg. Br. 1995, 77, 677-683.
  7. Rodeo, S.A.; Hannafin, J.A.; Tom, J.; Warren, R.F.; Wickiewicz, T.L. Immunolocalization of cytokines and their receptors in adhesive capsulitis of the shoulder. J. Orthop. Res. 1997, 15, 427-436.

Point 10: The number and quality of tables and figures are appropriate to transmit the main information of the paper.

Response 10: Thanks for the kind comments.

Round 2

Reviewer 2 Report

The authors answered to my comments properly: well done!